# Lymphovascular Space Invasion in Early-Stage Endometrial Cancer (LySEC): Patterns of Recurrence and Predictors. A Multicentre Retrospective Cohort Study of the Spain Gynecologic Oncology Group

**DOI:** 10.3390/cancers15092612

**Published:** 2023-05-04

**Authors:** M. Reyes Oliver-Perez, Pablo Padilla-Iserte, Octavio Arencibia-Sanchez, Cristina Martin-Arriscado, Juan Carlos Muruzabal, Berta Diaz-Feijóo, Silvia Cabrera, Pluvio Coronado, M. Belen Martín-Salamanca, Manuel Pantoja-Garrido, Josefa Marcos-Sanmartin, Elena Cabezas-López, Cristina Lorenzo, Duska Beric, Jose Ramon Rodriguez-Hernandez, Fernando Roldan-Rivas, Juan Gilabert-Estelles, Lourdes Sanchez, Maria Laseca-Modrego, Carmen Tauste-Rubio, Blanca Gil-Ibañez, Alvaro Tejerizo-Garcia

**Affiliations:** 1Gynecologic Oncology Unit, Department of Obstetrics and Gynecology, Hospital Universitario 12 de Octubre, 12 de Octubre Research Institute (i+12), 28041 Madrid, Spain; 2Department of Gynaecologic Oncology, La Fe University and Polytechnic Hospital, 46026 Valencia, Spain; 3Department of Gynecologic Oncology, University Hospital Materno-Infantil de Canarias, 35016 Las Palmas de Gran Canaria, Spain; 4Scientific Support Unit, Hospital Universitario 12 de Octubre, 12 de Octubre Research Institute (i+12), 28041 Madrid, Spain; 5Department of Gynecologic Oncology, Complejo Hospitalario de Navarra, 31008 Pamplona, Spain; 6Institute Clinic of Gynecology, Obstetrics and Neonatology, Hospital Clinic de Barcelona, Institut d’Investigacions Biomèdiques August Pi I Sunyer (IDIBAPS), Universitat de Barcelona, 08036 Barcelona, Spain; 7Gynecologic Oncology Unit, Gynecology Department, Hospital Universitari Vall d’Hebron, Universitat Autònoma de Barcelona, 08035 Barcelona, Spain; 8Women’s Health Institute of the Hospital Clínico San Carlos, IdISSC, School of Medicine, Complutense University Madrid, 28040 Madrid, Spain; 9Department of Gynecology, Hospital Universitario de Getafe, 28905 Madrid, Spain; 10Department of Gynecology and Obstetrics, University Hospital Virgen Macarena, 41009 Sevilla, Spain; 11Departments of Obstetrics and Gynecology, Dr. Balmis General University Hospital, 03010 Alicante, Spain; 12Department of Public Health, Miguel Hernandez University, Sant Joan D’Alacant, 03550 Alicante, Spain; 13Institute for Health and Biomedical Research (ISABIAL), 03010 Alicante, Spain; 14Department of Gynecologic Oncology, University Hospital Ramón y Cajal, 28034 Madrid, Spain; 15Department of Obstetrics and Gynecology, Hospital Nuestra Señora de la Calendaria, 38010 Santa Cruz de Tenerife, Spain; 16Department of Obstetrics and Gynecology, Hospital Universitario de Torrevieja, 03186 Alicante, Spain; 17Department of Gynecology and Obstetrics, University Hospital Virgen de La Arrixaca, El Palmar, 30120 Murcia, Spain; 18Department of Obstetrics and Gynaecology, Clinico Lozano Blesa Hospital, 50009 Zaragoza, Spain; 19Department of Pediatrics, Obstetrics and Gynecology, University General Hospital of Valencia, 46014 Valencia, Spain; 20Department of Gynecology and Obstetrics, University General Hospital of Ciudad Real, 13005 Ciudad Real, Spain

**Keywords:** endometrial cancer, recurrence, survival, LVSI, lymphovascular space invasion

## Abstract

**Simple Summary:**

In patients with early-stage endometrioid endometrial cancer, the presence of lymph vascular space involvement (LVSI) correlates with nodal metastases, shorter disease-free survival and overall survival. However, the effect of LVSI on recurrence patterns of these patients has been poorly studied, and the optimal adjuvant treatment remains unclear. Additionally, positive LVSI is indicative for nodal assessment, however, this parameter is usually not Known until a final pathology report. The main aim of our study was to analyze oncological outcomes and patterns of recurrence of these patients according to LVSI status, as well as to determine preoperative predictors of positive LVSI. We confirmed in a large multi-institutional cohort of patients (3546 participants), that positive LVSI is an independent prognostic factor for distant recurrences (HR 2.37) but not for local recurrence. In addition, deep myometrial invasion, high-grade tumors, cervical stroma invasion, and tumor diameter ≥ 2 cm are independent predictors of positive LVSI.

**Abstract:**

The main aim is to compare oncological outcomes and patterns of recurrence of patients with early-stage endometrioid endometrial cancer according to lymphovascular space invasion (LVSI) status. The secondary objective is to determine preoperative predictors of LVSI. We performed a multicenter retrospective cohort study. A total of 3546 women diagnosed with postoperative early-stage (FIGO I-II, 2009) endometrioid endometrial cancer were included. Co-primary endpoints were disease-free survival (DFS), overall survival (OS), and pattern of recurrence. Cox proportional hazard models were used for time-to-event analysis. Univariate and multivariate logistical regression models were employed. Positive LVSI was identified in 528 patients (14.6%) and was an independent prognostic factor for DFS (HR 1.8), OS (HR 2.1) and distant recurrences (HR 2.37). Distant recurrences were more frequent in patients with positive LVSI (78.2% vs. 61.3%, *p* < 0.01). Deep myometrial invasion (OR 3.04), high-grade tumors (OR 2.54), cervical stroma invasion (OR 2.01), and tumor diameter ≥ 2 cm (OR 2.03) were independent predictors of LVSI. In conclusion, in these patients, LVSI is an independent risk factor for shorter DFS and OS, and distant recurrence, but not for local recurrence. Deep myometrial invasion, cervical stroma invasion, high-grade tumors, and a tumor diameter ≥ 2 cm are independent predictors of LVSI.

## 1. Introduction

Endometrial cancer is the most frequent disease of the female genital tract in developed countries, with a relatively favorable prognosis [1]. However, approximately 20% of women with early-stage endometrial cancer will have a recurrence with a consequent lower overall survival (OS) [2]. As such, identifying risk factors for recurrence is crucial, specifically in those patients for whom adjuvant treatment is not always recommended or when all factors of that decision are not Known.

In this regard, lymphovascular space invasion (LVSI), defined as the presence of tumor cells within endothelial-lined channels outside the main tumor (Figure 1), has been postulated as one of the first steps in the metastatic spread of endometrial cancer [1,2,3,4]. Positive LVSI is significantly correlated with nodal metastases, shorter disease-free survival (DFS) and OS (1–8). For this reason, patients with low-risk features and positive LVSI are upgraded to the high-intermediate risk class for recurrence, and complete surgical staging with pelvic and paraaortic lymphadenectomy and/or adjuvant treatment is recommended [5,6]. Unfortunately, usually it is not possible to know LVSI status until a final pathology report. Even in that setting, LVSI diagnosis is linked to an observer’s experience [7]; yet there is a low rate of agreement between observers (kappa coefficient = 0.3) [7] and only moderate accuracy for LVSI on a frozen section (68.3–92.4%) [8,9].

To the best of our knowledge, only a few modest studies have evaluated the effect of LVSI on recurrence patterns in patients with early-stage endometrioid endometrial carcinoma, while the optimal adjuvant treatment strategy for patients with positive LVSI remains unclear [1,4,10,11,12]. In addition, few authors have analyzed predictors of LVSI, which could become a useful tool for correct lymph node surgical planning in patients with early-stage endometrioid endometrial cancer [13,14,15,16]. The main objective of this multi-institutional study was to analyze survival outcomes and patterns of recurrence in a cohort of women with early-stage endometrioid endometrial cancer with and without LVSI. Secondary objective was to investigate factors possibly associated with LVSI and to determine which of those factors could act as preoperative predictors of LVSI.

## 2. Materials and Methods

### 2.1. Study Design, Setting, and Oversight

We present a multicenter retrospective study endorsed by the Spain Gynecologic Oncology Group. The study was conducted in accordance with the Declaration of Helsinki. All researchers agreed to treat the data confidentially in accordance with the European General Data Protection Regulation [17]. The protocol and all amendments were approved by institutional review boards or ethics committees of each participating institution (N° CEIm: 21/019), which waived informed consent from patients due to the retrospective nature of the study. 

### 2.2. Cohort Selection and Study Variables 

We selected patients diagnosed with postoperative endometrioid endometrial cancer confined to the uterus (International Federation of Gynaecology and Obstetrics, FIGO, 2009 stage I–II) [18] and who had received at least a hysterectomy and bilateral salpingo-oophorectomy as surgical treatment. Lymph node assessment was performed based on preoperative risk factors for lymph node metastases according to European guidelines (high tumor grade (G3), myometrial invasion ≥ 50% and/or cervical stromal invasion) [5]. Performing sentinel lymph node biopsy or/and pelvic lymphadenectomy (with or without paraaortic lymphadenectomy) depended on protocol of each center. Exclusion criteria were non-endometrioid histology, synchronous tumors, FIGO 2009 stage III–IV [18], and unknown LVSI status. Extrauterine involvement was assessed preoperatively using imaging procedures such as pelvic magnetic resonance imaging and/or computed tomography or a positron emission tomography scan. 

Patients were divided into two groups depending on their positive or negative LVSI statuses. Based on the ESGO/ESTRO/ESP definition of prognostic risk groups, LVSI was considered positive when a diffuse or multifocal presence of tumor cells inside a space surrounded by endothelial cells was detected on hematoxylin-eosin-stained sections (substantial LVSI). No or focal LVSI was considered negative [4,5]. Tissue pathology was reported by the pathologists of each center, which, in all cases, specialized in gynecological oncology.

Demographic and clinico-pathological data were extracted the medical record (clinic and operative notes, radiologic and pathologic reports). The surgical variables collected were surgical staging procedure and surgical approach. Tumor grade was reported according to World Health Organization’s classification (1988) [19]; depth of myometrial invasion was recorded as an invasion less or equal to/more than half of the thickness of the myometrium; and maximum tumor diameter was determined as the largest of the three macroscopic measurements of tumor size. As FIGO stage changed during the study period, all patients were reclassified according to FIGO 2009 [18]. Finally, data on adjuvant treatment (vaginal brachytherapy, external beam radiation, and chemotherapy scheme); time of follow-up; time to recurrence or death; and location of recurrence were collected. 

### 2.3. Outcomes

Coprimary endpoints were DFS, OS and pattern of recurrence. DFS was defined as time from the date of surgery to the date of first recurrence. OS was calculated from the date of surgery to the date of death due to any cause. Finally, pattern of recurrence was defined according to the first site of recurrence. Those recurrences limited to the pelvic area and vaginal vault were considered local, while abdominal recurrences outside the pelvic area (peritoneal carcinomatosis, distant metastasis, and para-aortic lymph nodal metastases) were considered distant metastasis. 

### 2.4. Statistical Analysis

Categorical variables were expressed as absolute and relative frequency. Continuous variables were expressed as mean (standard deviation; SD) and median (interquartile range; IQR) according to a normality test (Kolmogorov–Smirnov test). For the estimation of differences between variables, Chi-square with Bonferroni adjustment when needed, a Student’s *t*-test if parametric, or the non-parametric Mann–Whitney U test was used.

OS and DFS were estimated using the Kaplan–Meier method and described by median and range. Differences in OS and DFS between groups were tested using the log-rank test. A Cox proportional hazards model was fitted to estimate hazard ratio (HR) and the corresponding 95% confidence interval (CI). A multivariable model was created with all confounding and relevant factors and had a *p*-value of <0.1 in the univariate analysis. The best multivariable statistical model was selected using Akaike Information Criterion.

A multivariate logistic regression model was used to analyze the association between the presence of LSVI and the potentially confounding variables that reached a significance level of <0.1 in different univariate models. Results were presented as odds ratio (OR) and 95% confidence interval (CI). The best multivariable statistical model was selected using the Akaike Information Criterion. Factors associated with the presence of LVSI, and distant recurrence were estimated using competing risks regression in DFS, considering the different existing risks. The Weibull function was used. A Cox proportional hazards model was fitted to estimate HR and the corresponding 95% CI, and the model represented cumulative risk during the follow-up period.

All analyses were done using Stata InterCooled for Windows version 16 (Stata Statistical Software Release 16, StataCorp LLC, College Station, TX, USA) and a significance level of two-tailed *p* < 0.05.

## 3. Results

### 3.1. Study Population

The data included in this study was collected from 16 Spanish centers. A total of 4958 patients with endometrial cancer who underwent surgery were studied. Of them, 3546 patients met the eligibility criteria (Appendix A). Patients were divided into two groups based on the LVSI status: 518 (14.6%) with positive LVSI and 3028 (85.4%) with negative LVSI. Clinicopathologic features of all patients are shown in Table 1.

A total of 1505 patients received some adjuvant treatment (42.4%), 1446 radiotherapy alone (40.8%), and 59 chemotherapy alone or with radiotherapy (1.6%). Adjuvant treatment was administered mor frequently in patients with positive LVSI than in those with negative LVSI (83.4% vs. 35.4%, *p* < 0.001) (Table 1). 

### 3.2. Survival Analysis According to LVSI Status 

At the data cutoff, this analysis (1 November 2022) median follow-up duration was 43.74 months (range: 24.00 to 63.31 months). Estimated DFS and OS at 5 years were, respectively, 91.9% (95% CI: 90.5–93.2) and 92.1% (95% CI 90.8; 93.2) for patients with negative LVSI versus 78.9% (95% CI: 74.1–82.8) and 79.0% (CI 95% 74.4; 82.9) for patients with positive LVSI (*p* < 0.001) (Figure 2).

In the univariate analysis, positive LVSI was a risk factor for recurrence (HR = 2.9; 95% CI: 2.2–3.8; *p* < 0.001). After adjustment for age ≥ 70 years, tumor diameter ≥ 2 cm, high-grade (G3), myometrial invasion ≥ 50%, and cervical stromal invasion (FIGO stage II), positive LVSI remained as risk factor for recurrence with a HR of 1.9 (95% CI: 1.3–2.5; *p* < 0.001) (Table 2). 

In addition, positive LVSI was a risk factor for death in the univariate analysis (HR = 2.5; 95% CI: 1.9–3.2; *p* < 0.001). After adjustment for age ≥ 70 years, tumor diameter ≥ 2 cm, high-grade (G3), myometrial invasion ≥ 50%, cervical stromal invasion (FIGO stage II), lymph node assessment, and adjuvant treatment, code as some adjuvant treatment, positive LVSI remained as risk factor for death with a HR of 2.1 (95% CI: 1.5–2.9; *p* < 0.001) (Table 2). 

### 3.3. Patterns of Recurrence

During the follow-up period, 278 patients (7.8%) recurred. Of these patients, 92 were in the positive LVSI group (17.8%) and 186 (6.1%) were in the negative LVSI group (*p* < 0.001) (Table 1 and Figure 2). Regarding the localization of recurrence, recurrences in the vaginal vault and limited to the pelvis were more frequent in patients with negative LVSI (38.2% and 21.7%, respectively), while distant were more common in those with positive LVSI (61.3% and 78.2%, respectively) (*p* = 0.01) (Table 3). 

In the univariate analysis, positive LVSI was a risk factor for distant recurrence (HR = 3.9; 95% CI: 2.9–5.3; *p* < 0.001), but not for local recurrence (HR = 1.5; 95% CI: 0.9–2.6; *p* = 0.112). After adjustment for age ≥ 70 years, high-grade (G3), myometrial invasion ≥ 50%, and adjuvant treatment, positive LVSI remained an independent risk factor for distant recurrence with a HR of 2.4 (95% CI: 1.7–3.4, *p* < 0.001) (Figure 2 and Table 2).

### 3.4. Predictors of LVSI

In the univariate analysis, prevalence of positive LVSI was greater in those with older age (*p* < 0.001), lower body mass index (*p* = 0.027), tumor diameter ≥ 2 cm (*p* < 0.001), myometrial invasion ≥ 50% (*p* < 0.001), high-grade (G3) (*p* < 0.001), and in those patients with cervical stromal invasion (*p* < 0.001) (Table 4). Multivariate analysis revealed that tumor diameter ≥ 2 cm (OR 2.03 95% CI 1.45; 2.85), deep myometrial invasion (OR 3.04 95% CI 2.36; 3.92), high grade tumors (OR 2.54 95% CI 1.84; 3.5), and cervical stroma invasion (OR 2.01 95% CI 1.34; 3.02), remained as independent risk factors for positive LVSI (Table 4). 

## 4. Discussion

### 4.1. Main Findings

The results of this large multi-center study indicate that positive LVSI is an independent risk factor for shorter DFS (HR = 1.8; *p* < 0.001) and OS (HR = 2.1; *p* < 0.001) in patients with early-stage endometrioid endometrial cancer, with a high prevalence of distant recurrences in patients with positive LVSI compared to patients with negative LVSI. Furthermore, positive LVSI is an independent risk factor (HR = 2.4; *p* < 0.001) for distant recurrence but not for local recurrence. Deep myometrial invasion (OR = 3.04; *p* < 0.001), high-grade tumors (OR = 2.54; *p* < 0.001), cervical stroma invasion (OR = 2.03; *p* = 0.001) and/or tumor size ≥ 2 cm (OR = 2.03; *p* < 0.001) are independently associated with a high risk of positive LVSI.

### 4.2. Results in the Context of Published Literature

Similar to our results, the presence of LVSI has been described in the literature as occurring in approximately 15% of FIGO stage I–II endometrioid endometrial cancer [1,2,4,10]. This parameter has been significantly associated to pelvic and para-aortic lymph node metastases [2,4,10,20,21,22]. However, LVSI is more than a surrogate for lymph node spread. It has been shown to be an independent prognostic factor, with shorter rates of DFS and OS even in patients with early-stage node-negative disease [1,3,4,5,10]. Consistently, in our study, positive LVSI was significantly associated with decreased DFS (HR = 1.9; *p* < 0.001) and OS (HR = 2.1; *p* < 0.001), even after adjusting for the main prognostic factors. Specifically, substantial LVSI, defined as diffuse or multifocal presence of tumor cells inside a space surrounded by endothelial, in contrast to focal or no LVSI, is the strongest independent prognostic factor for recurrence and OS [4]. Consequently, in stage I endometrial cancer patients with substantial LVSI, regardless of the degree and depth of myometrial invasion, complete surgical staging and/or adjuvant treatment is recommended [5,6]. Although, in recent years, molecular endometrial cancer classification has prompted a paradigm shift towards a model based on molecular features, proving to be one of the most robust prognostic values, even when a molecular profile is known and applied, LVSI remains an independent risk factor for DFS and OS [23,24,25]. Consequently, integrated models based on molecular features and histology are preferable. In these models, LVSI remains one of the main prognostic factors [6,24].

However, to date, only a few studies have focused on the patterns of recurrence of patients with early-stage endometrioid endometrial cancer and positive LVSI, and the optimal adjuvant treatment strategy for these patients remains unclear [1,4,10,11,12]. Similar to our results, most of these studies reported strikingly high distant recurrences in patients with positive LVSI compared to patients with negative LVSI [1,11,12]. Furthermore, in our cohort, positive LVSI was an independent risk factor for distant recurrence (HR = 2.4; *p* < 0.001) but not for local recurrence. Critically, the high rate of adjuvant radiotherapy administered to patients with positive LVSI in our cohort (83.4% vs. 35.4% for patients with negative LVSI) may have improved the local control in positive LVSI patients without influencing distant recurrence rate. On this point, Bosse et al. reported that positive LVSI was not predictive of local recurrence when adjusted for radiotherapy received; but radiotherapy had no impact on the risk of distant metastasis and OS (4). Therefore, non-local recurrences seem to be the most serious adverse risk factors to survival in these patients, which focuses on optimization of postoperative systemic treatment. However, there is no data showing a specific benefit of chemotherapy for these patients [4,5,26]. In this regard, the results of the ongoing PORTEC-4a clinical trial will provide prospective data on which factors may be used for selecting patients, including those with positive LVSI, as candidates for systemic therapies [23,24]. 

However, until all these issues are clarified, LVSI will remain as an indicative of lymph node assessment [5,6]. Also, despite the integration of molecular classification, LVSI will probably persist as one of the main risk factors for nodal involvement in patients with endometrial cancer [23,24,25]. As LVSI is generally not available until final histology report [8,9], identifying preoperative predictors of LVSI has become essential in surgical nodal assessment planning in patients with preoperative endometrioid endometrial cancer confined to the uterus. On this point, in our cohort of patients, deep myometrial invasion, a high tumor grade, cervical stromal involvement, and/or a tumor diameter ≥ 2 cm were independently associated with positive LVSI. Although, the first three markers are indicative of nodal assessment in patients with endometrioid endometrial cancer, tumor size is still not mandatory [5]. Based on our results, a tumor diameter ≥ 2 cm could play an important role in lymph node surgical planning of patients with preoperative low risk endometrioid endometrial cancer (myometrial invasion < 50%, a low tumoral grade, and non-cervical involvement). Tumor diameter could be accurately estimated preoperatively by imaging techniques [27,28]. Therefore, we suggest that in patients with tumor diameters ≥ 2 cm in preoperative settings, regardless of the depth of myometrial invasion or histological grade, nodal evaluation should be always considered. 

### 4.3. Strengths and Limitations 

The main limitation of this study is its retrospective nature and potential confounding factors due to the non-homogeneity of the population (nodal staging and adjuvant treatment are not standardized). Despite these limitations, our research represents one of the largest (3546 participants) multi-institutional cohorts of patients, reflecting the real-life management of patients with surgically treated endometrioid endometrial carcinoma. In addition, the long follow-up periods, which provide reliable data on recurrence and survival, should be noticed. Finally, pathological evaluation was performed by pathologists specialized in gynecological oncology in all cases, ensuring highly accurate reports. 

### 4.4. Implications for Future Research and Practice

Future research should be directed towards evaluating the relationship between LVSI status and molecular subclasses of endometrial cancer, to identify patients who would benefit from adjuvant—especially systemic- therapy which could prevent distant recurrences in patients with early-stage endometrioid endometrial cancer. However, until molecular classification is systematically implemented in all centers, pathologic features remain crucial in surgical planning and adjuvant treatment. In this context, LVSI is one of the strongest predictors of nodal metastasis and it is indicative of nodal assessment, even in the absence of other risk pathologic factors. However, LVSI status is usually not known until a final pathology report. In this context, one of the most important implications for practice of our study is that tumor diameter ≥ 2 cm is independently associated with a high risk of positive LVSI. This result could play an important role in surgical nodal planning, especially in patients with preoperative low-risk endometrial cancer. 

## 5. Conclusions

In early-stage endometrioid endometrial cancer, positive LVSI is an independent risk factor for shorter DFS and OS. Distant recurrences are more frequent in patients with positive LVSI compared to patients with negative LVSI. LVSI is an independent risk factor for distant recurrence but not for local recurrence. Deep myometrial invasion, cervical stroma invasion, high-grade tumors, and a tumor diameter ≥ 2 cm are independent risk factors for positive LVSI. In preoperative low-risk patients, a tumor size ≥ 2 cm could have an impact on surgical planning; nodal assessment in these patients should be considered. 

## Figures and Tables

**Figure 1 cancers-15-02612-f001:**
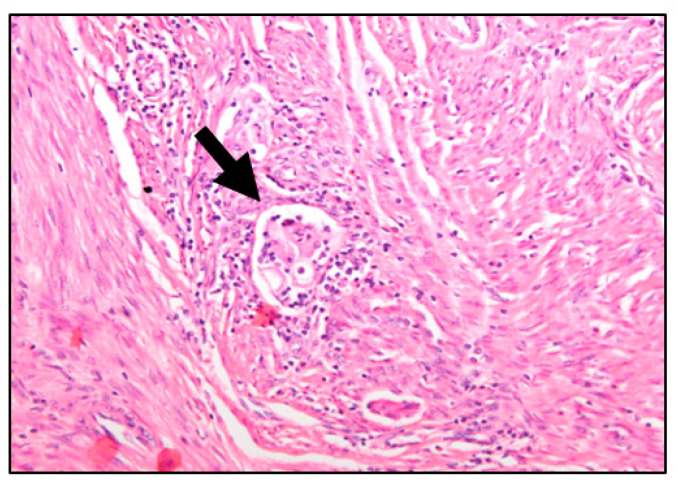
Hematosylin-eosin-stained section from an endometriosis endometrial tumor (magnification 20×). Lymphovascular space invasion can be observed (arrow): Cohesive tumor cells within a space surrounded by endothelial cells.

**Figure 2 cancers-15-02612-f002:**
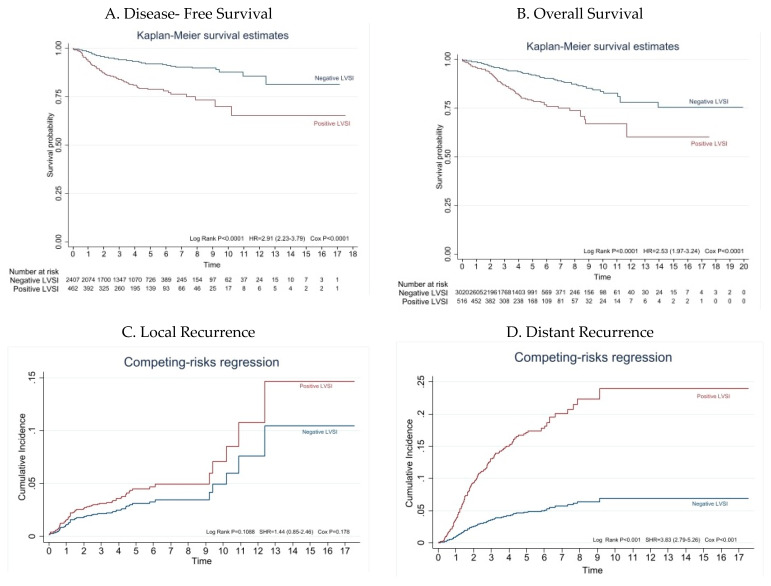
Survival analysis competing-risk regression for recurrence. (**A**,**B**) Kaplan–Meier analysis of DFS and OS for patients with early-stage endometrioid endometrial cancer in the positive LVSI group (red line) and negative LVSI group (blue line). The figure represents disease-free survival (**A**) and overall survival (**B**–**D**) Competing-risk regression for recurrence in patients with early-stage endometrioid endometrial cancer in the positive LVSI group (red line) and negative LVSI group (blue line). The figure represents risk of local recurrence (**C**) and distant recurrence (**D**). Local recurrences were defined as all those limited to the pelvic area and vaginal vault Abdominal recurrences outside the pelvic area (peritoneal carcinomatosis, distant metastasis and para-aortic lymph nodal metastases) were considered distant metastases. LVSI, lymphovascular space involvement.

**Table 1 cancers-15-02612-t001:** Clinicopathological characteristics of all patients and stratified according to LVSI-status.

Characteristic	Total (*n* = 3546)	Negative LVSI (*n* = 3028)	Positive LVSI (*n* = 518)	*p*-Value
Age (yr)				
<70	2362 (66.7)	2043 (67.5)	319 (61.6)	
≥70	1181 (33.3)	982 (32.5)	199 (38.4)	0.008
NR	3	3	0	
BMI				
Median (IQR)	31.2 (26.7–35.9)	31.2 (26.7–36.1)	30.5 (26.1–35.0)	0.042
NR	634	559	75	
Surgical approach				
MIS	2699 (76.2)	2287 (75.6)	412 (79.5)	0.054
Laparotomy	844 (23.8)	738 (24.4)	106 (20.5)	
NR	3	3		
LN assessment ^a^	1465 (41.3)	1156 (38.2)	309 (59.6)	<0.001
Tumour diameter (cm) ^b^				
<2	693 (26.4)	645 (28.8)	48 (12.3)	
≥2	1934 (73.6)	1592 (71.2)	342 (87.7)	<0.001
NR	919	791	128	
Grading ^c^				
Low grade (G1, G2)	3175 (89.7)	2786 (92.2)	389 (75.1)	
High grade (G3)	364 (10.3)	235 (7.8)	129 (24.9)	<0.001
NR	7	7	0	
MI ≥ 50%	973 (27.4)	693 (22.9)	280 (54.1)	<0.001
FIGO stage ^d^				
I	3308 (93.3)	2865 (94.6)	443 (85.5)	<0.001
II	238 (6.7)	163 (5.4)	75 (14.5)	
Adjuvant treatment				
None	2041 (57.6)	1955 (64.6)	86 (16.6)	
BT alone	708 (20.0)	544 (18.0)	164 (31.7)	
EBRT+/−BT	738 (20.8)	495 (16.3)	243 (46.9)	<0.001
CT alone or with RT	59 (1.6)	34 (1.1)	25 (4.8)	
Recurrence				
No	3268 (92.2)	2842 (93.9)	426 (82.2)	<0.001
Yes	278 (7.8)	186 (6.1)	92 (17.8)	
Recurrence ^e^				
No	3268 (92.2)	2842 (93.9)	426 (82.2)	<0.001
Local and vaginal vault	92 (2.5)	72 (2.3)	20 (3.8)	
Distant metastases	186 (5.3)	114 (3.8)	72 (14.0)	

Data presented as number (percentage, %) or Mean (standard deviation, SD) or median (Interquartile range, IQR: p25; p75). ^a^ LN assessment: sentinel lymph node biopsy (with bilateral migration) and/or pelvic lymphadenectomy; ^b^ Maximum tumor diameter was determined as the largest of the three macroscopic measurements of tumor size; ^c^ According to World Health Organization’s classification (1988) [19]; ^d^ FIGO 2009 [18]. ^e^ Local recurrences were defined as all those limited to the pelvic area and vaginal vault, and abdominal recurrences outside the pelvic area (peritoneal carcinomatosis, distant metastasis, and para-aortic lymph nodal metastases) were considered distant metastases. NR, not reported; LVSI, lymphovascular space invasion; BMI, body mass index; MIS, minimal invasive surgery; LN, lymph node; MI, myometrial invasion; BT, brachytherapy; EBRT, External beam radiation; CT, chemotherapy; RT, radiotherapy.

**Table 2 cancers-15-02612-t002:** Univariate and multivariate Cox regression analysis of select covariates for DFS, OS, distant, and local recurrence.

Characteristics	DFS				OS			
	Univariate		Multivariate		Univariate		Multivariate	
	HR (95%CI)	*p*-Value	HR (95% CI)	*p*-Value	HR (95%CI)	*p*-Value	HR (95% CI)	*p*-Value
LVSI present	2.9 (2.2–3.8)	<0.001	1.9 (1.3–2.5)	<0.001 ^f^	2.5 (1.9–3.2)	<0.001	2.1 (1.5–2.9)	<0.001 ^g^
Age ≥ 70 y.	1.6(1.3–2.1)	<0.001	1.6 (1.2–2.1)	<0.002	3.80 (3.0–4.8)	<0.001	3.6 (2.7–4.7)	<0.001
Tumour diameter ≥ 2 cm ^a^	2.4(1.6–3.6)	<0.001	1. 7 (1.1–2.6)	0.015	2.1 (1.5–3.1)	<0.001	1.63 (1.1–2.4)	0.012
High grade ^b^	3.4 (2.5–4.5)	<0.001	2.6 (1.8–3.5)	<0.001	2.4 (1.8–3.1)	<0.001	2.5 (1.8–3.5)	<0.001
MI ≥ 50%	2.1 (1.6–2.7)	<0.001	1.3 (0.9–1.8)	0.066	1.7 (1.3–2.1)	<0.001	1.5 (1.0–2.0)	0.026
Figo Stage II ^c^	2.8 (1.9–3.9)	<0.001	1.8 (1.2–2.7)	0.007	1.7 (1.2–2.4)	0.007	1.4 (0.9–2.3)	0.124
LN assessment ^d^	1.1 (0.9–1.5)	0.309	-	-	0.7 (0.5–0.9)	0.0009	0.5 (0.3–0.6)	<0.001
Adjuvant treatment ^e^	2.1 (1.6–2.7)	<0.001	-	-	1.2 (0.9–1.5)	0.105	0.6 (0.4–0.8)	0.002
	**Distant Recurrence ^h^**				**Local Recurrence ^h^**			
	**Univariate**		**Multivariate**		**Univariate**		**Multivariate**	
	**HR (95%CI)**	***p*-Value**	**HR (95% CI)**	***p*-Value**	**HR (95%CI)**	***p*-Value**	**HR (95% CI)**	***p*-Value**
LVSI present	3.9 (2.8–5.3)	<0.001	2.4 (1.7–3.4)	<0.001 ^i^	1.5 (0.9–2.6)	0.112	-	-
Age ≥ 70 y.	1.6 (1.1–2.2)	0.005	1.4 (1.0–1.9)	0.037	1.7 (1.0–2.7)	0.019	1.8 (1.1–2.9)	0.022
Tumour diameter ≥ 2 cm ^a^	2.4 (1.5–4.1)	0.001	-	-	2.3 (1.1–4.6)	0.022	2.3 (1.1–4.8)	0.020
High grade ^b^	4.1 (2.9–5.7)	<0.001	2.6 (1.8–3.7)	<0.001	2.2 (1.2–3.9)	0.006	3.1 (1.6–5.9)	0.001
MI ≥ 50%	2.7 (1.9–3.7)	<0.001	1.5 (1.0–2.1)	0.034	1.3 (0.8–2.1)	0.26	-	-
LN assessment ^d^	1.5 (1.1–2.1)	0.007	-	-	0.6 (0.4–0.9)	0.029	-	-
Adjuvant treatment ^e^	3.5(2.5–5.1)	<0.001	1.6 (1.0–2.6)	0.031	0.9 (0.6–1.4)	0.519	-	-

^a^ Maximum tumor diameter was determined as the largest of the three macroscopic measurements of tumor size; ^b^ According to World Health Organization’s classification (1988). High grade is considered grade 3 [19]; ^c^ FIGO 2009 [18]; ^d^ LN assessment: sentinel lymph node biopsy (with bilateral migration) and/or pelvic lymphadenectomy; ^e^ Coded as some adjuvant treatment (RT+/−CT) vs. none; ^f^ Value adjusted for age, tumor diameter, high grade, MI ≥ 50%, and FIGO stage; ^g^ Value adjusted for age, tumor diameter, high grade, MI ≥ 50%, FIGO stage, LN assessment and adjuvant treatment; ^h^ Abdominal recurrences outside the pelvic area (peritoneal carcinomatosis, distant metastasis and para-aortic lymph nodal metastases) were considered distant metastases. Local recurrences were defined as all those limited to the pelvic area and vaginal vault; ^i^ Value adjusted for age, tumor diameter, high grade, MI ≥ 50%, and adjuvant treatment. DFS, Disease-free survival; OS, Overall survival; HR, Hazard ratio; CI, confidence interval; LVSI, lymphovascular space invasion; MI, myometrial invasion; LN, lymph node.

**Table 3 cancers-15-02612-t003:** Patients with recurrent disease. Patterns of recurrence.

Location	Total (*n* = 278)	LVSI Negative (*n* = 186)	LVSI Positive (*n* = 92)	*p*-Value
Local and Vaginal vault	92 (33.1)	72 (38.7)	20 (21.7)	0.012
Peritoneal carcinomatosis	44 (15.8)	31 (16.8)	13 (14.2)	
Metastatic lymph nodes	44 (15.8)	24 (12.9)	20 (21.7)	
Visceral metastases	98 (35.3)	59 (31.7)	39 (42.4)	
Local and vaginal vault ^a^	92 (33.1)	72 (38.7)	20 (21.8)	0.01
Distant metastases ^b^	186 (66.9)	114 (61.3)	72 (78.2)	

Data presented as number (percentage, %) ^a^ Local recurrences were defined as all those limited to the pelvic area and vaginal vault. ^b^ Abdominal recurrences outside the pelvic area (peritoneal carcinomatosis, distant metastasis, and para-aortic lymph nodal metastases) were considered distant metastases. LVSI, Lymphovascular space involvement.

**Table 4 cancers-15-02612-t004:** Univariate and multivariate analyses of clinicopathological characteristics in association to odds ratio to predict LVSI status.

Characteristics	Univariate Analysis		Multivariate Analysis	
	0 R (95%CI)	*p*-Value	0 R (95% CI)	*p*-Value
Age ≥70 years	1.29 (1.07–1.57)	0.008	0.83 (0.64–1.08)	0.17
BMI	0.98 (0.96–0.99)	0.027	0.98 (0.96–1.00)	0.054
Tumor diameter ≥ 2 cm ^a^	2.88 (2.10–3.95)	<0.001	2.03 (1.45–2.85)	<0.001
Myometrial invasion ≥ 50%	3.96 (3.26–4.80)	<0.001	3.04 (2.36–3.92)	<0.001
High grade (G3) ^b^	3.9 (3.09–4.99)	<0.001	2.54 (1.84–3.5)	<0.001
FIGO II ^c^	2.97 (2.22–3.98)	<0.001	2.01 (1.34–3.02)	0.001

^a^ Maximum tumor diameter was determined as the largest of the three macroscopic measurements of tumor size; ^b^ According to World Health Organization’s classification (1988) [19]; ^c^ FIGO 2009 [18]. OR, Odds ratio; CI, confidence interval; LVSI, lymphovascular space invasion; BMI, body mass ind.ex.

## Data Availability

All data relevant to the study are included in the article or uploaded as Appendix A.

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
