# Peer review of "Lymphovascular Space Invasion in Early-Stage Endometrial Cancer (LySEC): Patterns of Recurrence and Predictors. A Multicentre Retrospective Cohort Study of the Spain Gynecologic Oncology Group"

_cancers, 2023, doi:10.3390/cancers15092612_

Round 1

Reviewer 1 Report

 Lymphovascular Space Invasion in Early-Stage Endometrial 2 Cancer (LySEC): patterns of recurrence and predictors. A multi-3 centre retrospective cohort study of the Spain Gynecologic On-4 cology Group.

The manuscript is well written and once again establishes the prognostic significance of LVSI in endometrial cancer. They have analysed the data with appropriate statistical tools.
While acknowledging the uncertainty of adjuvant treatment of LVSI positive early-stage endometrial cancer patients, they do recommend criteria for lymph-node staging in these patients.
Here are some suggestions and a bit of clarification is needed.
Looking at the table 1, the incidence of LVSI positivity in G1 and 2 tumours is 12% and that in G3 is 35%. It would have been useful to analyse G1+G2 group separately to G3. The authors have recommended nodal staging in patients with >2cm tumours as well as those with >1/2 myometrial invasion irrespective of histology grade. Generally speaking, the ratio of LVSI to nodal involvement is 4:1 in G1& G2 tumours whereas it is 2.5:1 in G3 tumours. If they looked at the LVSI negative stage 1 and 2 and G1and G2 tumours they might find that in absence of LVSI in this group neither the myometrial invasion nor the tumour size would make any prognostic difference. I think a multivariable analysis in stage 1 and 2, LVSI negative and G1 and G2 histology the independent prognosis of tumour size and myometrial invasion would be informative in quantitating the risk of relapse and possible recommendation for adjuvant treatment.

Reviewer 2 Report

This research aims to investigate the impact of presence of LVSI in those individuals with early stage endometrioid endometrial cancer.  This is well written study that includes a large population of patients in a real world setting.  I have some questions for the authors to address.

1) The definition of LVSI positive include only those cases where LVSI was substantial.  If there was focal LVSI this was grouped into the LVSI negative group.

        a) What was the rationale behind including focal LVSI into the LVSI negative group?

        b) Was there any analysis done to look at the focal LVSI group?  If so, what were the implications of having focal LVSI?

2) In Table 1 under adjuvant therapy, it is noted the number of patients that received EBRT+/-BT.  Were there any patients that received BT alone as adjuvant therapy?  If so, were patterns of recurrence any different?  Based on PORTEC2 data, would expect many more individuals to have received BT without EBRT, so am surprised that this was not noted

3) In Table 2 there are multiple variables not reported in the multivariable analyses even though the predefined criteria of p <0.01 was met (eg adjuvant treatment in the DFS, tumor diameter and LN assessment in distant recurrence). What happened to these variables? 

4) Also in Table 2 there were variables that did not meet the predfined cutoff of p<0.01 and were included (eg adjuvant treatment in OS, age >70 and tumor diameter in local recurrence).  If a predefined cutoff is being set, then this should be used.  If exceptions are being made because of clinical reasons, then this needed to be stated in the manuscript.

Round 2

Reviewer 2 Report

I do not have any additional questions or concerns for the authors.